# Siglec receptors impact mammalian lifespan by modulating oxidative stress

**Flavio Schwarz[1,2,3†], Oliver MT Pearce[1,2,3†], Xiaoxia Wang[1,2,3], Annie N Samraj[1,2,3], Heinz Läubli[1,2,3], Javier O Garcia[4], Hongqiao Lin[5], Xiaoming Fu[5], Andrea Garcia-Bingman[1,3], Patrick Secrest[1,3], Casey E Romanoski[3], Charles Heyser[6], Christopher K Glass[3], Stanley L Hazen[5], Nissi Varki[1,7], Ajit Varki[1,2,3*], Pascal Gagneux[1,7*]**

[1]Glycobiology Research and Training Center, University of California, San Diego, San Diego, United States; [2]Department of Medicine, University of California, San Diego, San Diego, United States; [3]Department of Cellular and Molecular Medicine, University of California, San Diego, San Diego, United States; [4]Department of Psychology, University of California, San Diego, San Diego, United States; [5]Department of Cellular and Molecular Medicine, Cleveland Clinic Lerner Research Institute, Cleveland, United States; [6]Department of Neurosciences, University of California, San Diego, San Diego, United States; [7]Department of Pathology, Division of Comparative Pathology and Medicine, University of California, San Diego, San Diego, United States

*For correspondence: a1varki@ucsd.edu (AV); pgagneux@ucsd.edu (PG)

†These authors contributed equally to this work

**Abstract** Aging is a multifactorial process that includes the lifelong accumulation of molecular damage, leading to age-related frailty, disability and disease, and eventually death. In this study, we report evidence of a significant correlation between the number of genes encoding the immunomodulatory CD33-related sialic acid-binding immunoglobulin-like receptors (CD33rSiglecs) and maximum lifespan in mammals. In keeping with this, we show that mice lacking Siglec-E, the main member of the CD33rSiglec family, exhibit reduced survival. Removal of Siglec-E causes the development of exaggerated signs of aging at the molecular, structural, and cognitive level. We found that accelerated aging was related both to an unbalanced ROS metabolism, and to a secondary impairment in detoxification of reactive molecules, ultimately leading to increased damage to cellular DNA, proteins, and lipids. Taken together, our data suggest that CD33rSiglecs co-evolved in mammals to achieve a better management of oxidative stress during inflammation, which in turn reduces molecular damage and extends lifespan.

## Introduction

Aging is controlled partly by genetic factors, such as insulin/IGF-1, mTOR, AMPK, and Sirtuin signaling pathways (*Lopez-Otin et al., 2013*). Another important element affecting aging is thought to be cumulative damage to macromolecules by reactive oxygen and nitrogen species (ROS/RNS) induced by unbalanced cellular inflammatory responses, or generated via mitochondrial dysfunction (*Berlett and Stadtman, 1997*; *Dizdaroglu et al., 2002*). A large proportion of reactive oxygen species (ROS) formed in vivo is derived from the electron transport chain in mitochondria during cellular respiration. Additionally, ROS are generated in blood and tissue phagocytes upon release of superoxide radicals by NADPH oxidase in response to pathogens (*Finkel and Holbrook, 2000*). ROS can also be rapidly induced from resident local cells and recruited leukocytes upon tissue injury. Evolution towards an optimal trade-off between protective and damaging ROS levels in organisms includes the introduction of a number of enzymatic and non-enzymatic anti-oxidant mechanisms to maintain homeostasis and

**eLife digest** As we get older, we are more likely to become frail, be less mobile and develop heart disease, diabetes, and other age-related diseases. This is partly due to damage to tissues and organs that accumulates over the course of our lifetime. How quickly we age is controlled both by our genetics and by the environment we live in.

It is thought that damage to DNA, proteins, and other molecules in the body caused by chemically active molecules called reactive oxygen species (ROS) can influence aging. ROS are produced during respiration, immune responses, and other important processes in cells, but in excessive amounts they can be extremely harmful. To avoid damage to DNA and other important molecules, cells have several ways to control the levels of ROS.

One of the other hallmarks of aging is the development of chronic inflammation in tissues around the body, which is partly triggered by the immune system in response to cell damage. A group of genes called the *CD33rSIGLEC* genes are involved in controlling inflammation. The genomes of different mammal species carry different numbers of these genes, but it is not clear whether this alters the aging process in these animals.

In this study, Schwarz et al. investigated whether the *CD33rSIGLEC* genes influence the lifespans of mammals. Species with a higher number of *CD33rSIGLEC* genes generally have a longer lifespan than those with fewer of these genes. Mice that were missing one of these genes and were subjected to inflammation early in life showed signs of accelerated aging and had shortened lifespans compared with normal mice. As predicted, these mice also had higher levels of ROS, which led to a greater amount of damage to the DNA and other molecules in their bodies.

Schwarz et al.'s findings suggest that the *CD33rSIGLECs* co-evolved in mammals to help control the levels of ROS during inflammation, thereby reducing the damage to cells and extending the lifespan of the animals. Given that individual humans have different numbers of working *CD33rSIGLEC* genes, it would be interesting to see if this influences human lifespan.

mitigate damage. Accordingly, comparative studies have shown association between the longevity of a species and the capacity of cells in its individuals to resist oxidative stress (*Kapahi et al., 1999*; *Andziak et al., 2006*; *Brown and Stuart, 2007*). Understanding the finer details of this regulatory process might also provide access to measures for alleviating and controlling conditions associated with aging, a pressing medical challenge in a society with increasing lifespan.

In this study, we sought to determine whether the CD33rSiglecs impact aging and influence lifespan in mammals. Siglecs are mainly expressed by cells of the immune system and bind broadly to sialylated structures of the same cell or of neighboring cells through their extracellular domain (*Crocker et al., 2007*). Two classes of Siglecs are defined based on sequence homology and conservation. The first group (Sialoadhesin/Siglec-1, CD22/Siglec-2, MAG/Siglec-4 and Siglec-15) share low sequence identity but are conserved across mammals. In contrast, the genes encoding CD33rSiglecs underwent extensive rearrangements, including duplication, conversion, and pseudogenization, and therefore vary in number and in sequence between different mammal species (*Cao and Crocker, 2011*; *Padler-Karavani et al., 2014*; *Schwarz et al., 2015*). For instance, mice and humans (the two best studied organisms in this respect) express five and ten functional CD33rSiglecs, respectively (*Angata et al., 2004*). CD33rSiglecs in humans are numbered (e.g., Siglecs-3, -5, -6, -7, -8, -9, -10, -11, -XII, -14 and -16), while murine CD33rSiglecs (other than Siglec-3) are identified by a distinct alphabetical nomenclature (*Crocker et al., 2007*; *Macauley et al., 2014*). Although information regarding Siglec expression patterns is not comprehensive, it is known that many members are expressed in a cell type-specific manner. For instance, among the murine CD33rSiglecs, CD33 is expressed mainly in granulocytes, Siglec-E is expressed primarily in neutrophils, monocytes, microglia, and dendritic cells, Siglec-F is mainly found in eosinophils and mast cells, Siglec-G is predominantly expressed in B cells and some dendritic cells, and Siglec-H is primarily expressed in plasmacytoid dendritic cells (*Pillai et al., 2012*). Although it is not possible to identify clear CD33rSiglec orthologs between human and murine Siglecs, due to rapid Siglec evolution and deep divergence time between mice and humans, some Siglec receptors (for instance, Siglec-E and Siglec-9) are considered to be functional homologs (*Läubli et al., 2014*). Notably, there is no evidence so far for

a significant degree of functional redundancy among Siglecs. Despite the general low affinity of Siglecs towards the sialylated structures, it appears that each Siglec has unique sialoglycan specificity profile with regard to the type of sialic acid, its linkage and the composition of underlying glycan structure. Interestingly, CD33rSiglecs can transmit inhibitory signals into immune cells by phosphorylation of intracellular ITIM or ITIM-like domains, thus quenching pro-inflammatory cascades (*Crocker et al., 2007*). Recently, it has been shown that Siglecs can directly control Toll-like receptor (TLR) signaling by sustaining sialic acid-dependent interactions with TLRs and CD14 (*Chen et al., 2014*; *Ishida et al., 2014*). As the development of chronic inflammation is one of the hallmarks of aging (*Franceschi et al., 2005*), we investigated whether the number of CD33rSiglecs has co-evolved to modulate the aging process. We asked if the number of *CD33rSIGLEC* genes correlates with the maximum lifespan in mammalian species and found there was indeed a strong link, which was maintained after correction for phylogenetic or body mass constraints. We then tested if deletion of Siglec-E impacts longevity in mice. Indeed, Siglec-E-deficient mice exhibited accelerated signs of aging compared to littermate controls. We detected an increased rate of oxidative damage to cellular macromolecules at the systemic level, which we found to be related to a disrupted ROS homeostasis and early signs of aging. Finally, we tested the absence of Siglec-E in survival studies and found that these mice had significantly reduced longevity. Our combined data indicate that CD33rSiglecs regulate inflammatory damage and that the expansion of their number in the genome has coevolved with the extension of lifespan in mammals.

## Results

### The number of *CD33rSIGLEC* genes correlates with maximum lifespan

The evolutionary theory of germ line and disposable soma predicts that long-lived species assure their longevity through investments in more resilient somatic tissues (*Moore et al., 1991*; *Kirkwood, 1992*). It follows then that genes involved in management of cellular stress and repair of damage contribute to lifespan. Indeed, it has been experimentally shown that lifespan of eight mammalian species correlates to the ability of their primary fibroblasts to cope with stress (*Kapahi et al., 1999*). As Siglecs are capable of modulating cellular inflammatory responses and the number of genes encoding CD33rSiglecs varies widely between species (*Angata et al., 2004*), we asked if *CD33rSIGLEC* gene number correlates with maximum lifespan in mammals. A positive correlation was observed between these two parameters in the 14 mammalian species tested ($R^2 = 0.7630$) (*Figure 1A*, *Figure 1—figure supplements 1, 2*). As the genes encoding CD33rSiglecs are mainly found in a single syntenic cluster in each species, we considered the possibility that the observed correlation could be due to factors associated with the chromosomal environment surrounding these genes or due to hitchhiking effects with adjacent genes. We therefore examined the Kallikrein-related peptidase (*KLK*) gene cluster, which is located in the chromosomal region immediately adjacent to *CD33rSIGLECs* in most of the 14 species. Strikingly, the number of *KLK* genes showed a poor correlation with mammalian lifespan ($R^2 = 0.1825$) (*Figure 1B*). Next, we tested if the observed correlation of *CD33rSIGLEC*/maximum lifespan was due to a general expansion of genes encoding for cell surface receptors that interact with pathogens to initiate immune responses. Therefore, we examined Toll-like receptor (TLR) genes, which play important roles in innate recognition of PAMPs and DAMPs (*Janeway and Medzhitov, 2002*; *Beutler, 2009*), and genes for IgG Fc gamma receptors, which bind the Fc region of IgG to regulate immune responses (*Nimmerjahn and Ravetch, 2008*). Predicted gene numbers for both families showed only marginal association with maximum lifespan (*Figure 1C,D*).

Since closely related species may also share similar traits simply due to their common ancestry, data from different species may not be statistically independent. To control for such effects, we used phylogenetic comparative analysis using Phylogeny Generalized Least-Squares (PGLS) or Felsenstein's Independent Contrast (FIC) approaches. The correlation between CD33rSiglecs and longevity remained very strong after such phylogenetic correction (*Table 1* and *Figure 1—figure supplement 3*). Moreover, the correlation was maintained after mathematical correction for body mass represented by average adult body weight (*Table 2*), another factor known to correlate with metabolic rate and lifespan (*Manini, 2010*). Overall, a positive correlation was shown between the residual maximum lifespan and residual *CD33rSIGLEC* gene numbers (controlling for both body mass and phylogeny) (*Figure 1E,F*). Since the time that these data were originally collected and evaluated, additional genome sequences have become available. Therefore, in order to further test the strength of the correlation, we included

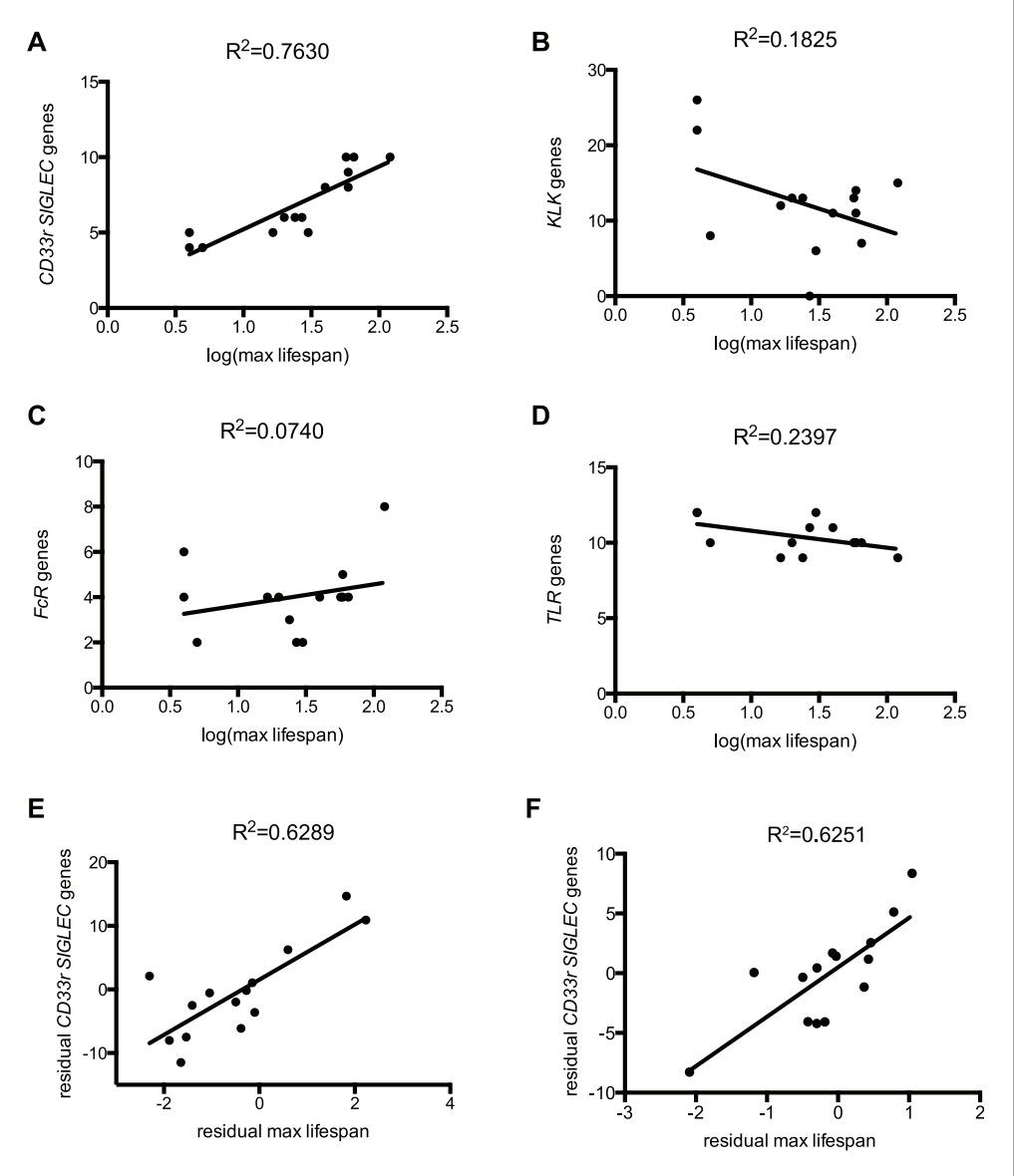

**Figure 1**. Correlation between gene numbers in gene families and maximum lifespan in mammals. Numbers of CD33rSiglecs (**A**), KLK (**B**), IgG Fc receptors (**C**), and TLRs genes (**D**) and maximum lifespan in 14 mammalian species listed in *Figure 1—figure supplement 1*. (**E** and **F**) Correlation of *CD33rSIGLEC*s and maximum lifespan after correction for average adult body weight and phylogeny. PGLS: λ = 1, phylogenetic tree I (**E**) or tree II (**F**) were used. The Pearson's correlation coefficient ($R^2$) for each plot is indicated.

The following figure supplements are available for figure 1:

**Figure supplement 1**. Data of 14 mammalian species used for analysis of correlation.

**Figure supplement 2**. Correlation between number of genes of each family and maximum lifespan.

**Figure supplement 3**. Phylogeny and tree branch information.

the data from three short-lived primate genomes (*Saimiri boliviensis*, *Tarsius syrichta*, and *Otolemur garnettii*). Interestingly, these genomes were found to have fewer *CD33rSIGLEC* genes (5, 5, and 4 genes, respectively). Addition of these data to the primary correlation did not change the statistical significance of the association between number of *CD33rSIGLEC* genes and maximum longevity

**Table 1.** Statistical analysis of the correlation between number of genes and maximum lifespan, corrected for phylogeny

| Gene family | PGLS | FIC |
| --- | --- | --- |
| Tree I | | |
| CD33rSIGLECs | 0.00016 | 0.00012 |
| KLKs | 0.49 | 0.17 |
| TLRs | 0.38 | 0.10 |
| IgG Fc receptors | 0.35 | 0.0016 |
| Tree II | | |
| CD33rSIGLECs | 0.00017 | 0.00011 |
| KLKs | 0.65 | 0.23 |
| TLRs | 0.32 | 0.14 |
| IgG Fc receptors | 0.39 | 0.0019 |

Phylogenetic comparative analysis was conducted in COMPARE 4.6b using Phylogeny Generalized Least-Squares (PGLS) or Felsenstein's Independent Contrast (FIC) approaches. Student's t-values were computed based on the regression slopes and the standard errors. Two-tailed probability (p) value of a Student's t-test was estimated using a degree of freedom of 11. The phylogenetic relationship of 14 mammalian species represented by Tree I and Tree II are indicated in *Figure 1—figure supplement 3*. Note that although FIC analysis obtained a significant p value for IgG Fc receptor gene family, the p value increased to 0.56 when human data was excluded. Thus, this correlation is driven by one outlier data point.

($R^2 = 0.661$ in logarithmic scale, $R^2 = 0.752$ in linear scale). Furthermore, based on the different sample size of different species tested, an adjusted value of maximum longevity of 90 years for humans was considered, in line with previous studies (*Lorenzini et al., 2005*). Notably, the overall correlation between number of *CD33rSIGLEC* genes and maximum longevity remained strong ($R^2 = 0.649$ in the logarithmic scale, $R^2 = 0.843$ in the linear scale).

Taken together, these data indicate that the number of *CD33rSIGLEC* genes correlates to lifespan in mammals. This correlation appears to be independent from phylogenetic constraints, from effects of genomic location, from a generally observed rapid evolution of receptors involved in immune responses and from body mass.

## Accelerated aging and reduced lifespan of Siglec-E-deficient mice

We decided to use a mouse model to seek experimental evidence for the observed correlation, as mice have a simplified CD33rSiglec profile compared to other mammalian model systems, in terms of number of genes and expression patterns. In fact, mice possess five CD33rSiglecs (namely, CD33, Siglec-E, -F, -G, -H). Among these, Siglec-E is the dominant receptor and it is strongly expressed on neutrophils, tissue macrophages (*Figure 2—figure supplement 1*), and microglia (*Zhang et al., 2004*; *Claude et al., 2013*). We monitored the survival of mice lacking Siglec-E (Siglec-E$^{-/-}$) over the course of 100 weeks in comparison to their control wild type littermates (WT) (*Figure 2A*). The survival study was carried out in two sequential cohorts totaling 117 WT and 120 Siglec-E$^{-/-}$ mice. Overall survival of the Siglec-E$^{-/-}$ males was markedly reduced compared to WT (48% and 70% remaining, respectively, when the experiment was terminated). Similarly, relative to the WT, the median survival of Siglec-E$^{-/-}$ females decreased by 17%. In an attempt to mimic natural conditions of early exposure to inflammatory insults, we exposed all groups of mice to a non-specific antigenic challenge early in life (heterologous cell membranes mixed with Freund's adjuvant). This treatment did not affect the viability of Siglec-E$^{-/-}$ mice over 100 weeks (*Figure 2—figure supplement 2*). No differences in general appearance or body weight were noted and no signs of specific pathologies were observed during the study (*Figure 2—figure supplement 3*). Hematological and biochemical analysis of blood samples at periodic intervals and at termination of the study did not reveal significant differences between the two groups (*Figure 2—figure supplement 4*). Additionally, there was no evidence indicative of systemic chronic disease, such as increased leukocyte counts, microcytic anemia, or hypoalbuminemia. Serum creatinine levels did not suggest diminished renal function. Higher values of alanine aminotransferase were noted for Siglec-E$^{-/-}$ animals but were not statistically different from the controls. Similarly, systematic histological analysis of multiple organs showed no evidence of pathological abnormalities, though we observed sporadic instances of periportal liver inflammation, and a slight increase in lung inflammation compared to control mice (*Figure 2—figure supplement 5*). Examination of kidneys showed that more of the Siglec-E$^{-/-}$ mice exhibited minor age-related glomerular changes, with thickening of glomerular tufts, visible on Periodic-Acid Schiff stains. These data were in line with previous work on the same mice at a younger age (*McMillan et al., 2013*).

We submitted the mice to a series of analyses to test if Siglec-E$^{-/-}$ animals exhibited exacerbated age-related defects. First, 80-week-old Siglec-E$^{-/-}$ mice showed a threefold increase in error rate in the Barnes maze test compared to controls (*Figure 2B,C*). These results are consistent with previous

**Table 2.** Statistical analysis of the correlation between number of genes and maximum lifespan, corrected for body weight

| Gene family | t value | p value |
|---|---|---|
| Tree I | | |
| CD33rSIGLECs | 3.521435 | 0.004786 |
| KLKs | 0.360601 | 0.725226 |
| TLRs | −0.701822 | 0.497370 |
| IgG Fc receptors | 1.656519 | 0.125834 |
| Tree II | | |
| CD33rSIGLECs | 3.660917 | 0.003748 |
| KLKs | 0.180227 | 0.860251 |
| TLRs | −0.726465 | 0.482726 |
| IgG Fc receptors | 1.589586 | 0.140235 |

Phylogenetic comparative analysis conducted in CAIC package. Average adult body weight and maximum lifespan of 14 mammalian species were log-transformed and phylogenetic regressions were run using pglmEstLambda in the CAIC package in R. This function uses the PGLS method and estimates λ with the average adult body weight controlled for. Student's t-values and two-tailed probability (p) values are shown. The phylogenetic relationship of 14 mammalian species represented by Tree I and Tree II are indicated in *Figure 1—figure supplement 3*.

reports of impairments in learning and in spatial memory in aged mice (*Kennard and Woodruff-Pak, 2011*). Deficits in the Barnes maze were not due to alterations in locomotor activity (*Figure 2—figure supplement 6*). Secondly, a blind test involving three independent observers noted increased hair graying in Siglec-E$^{-/-}$ males compared to WT (*Figure 2D*). Hair graying is related to incomplete maintenance of melanocyte stem cells through loss of the differentiated progeny that occurs physiologically during aging (*Nishimura et al., 2005*). After termination of the survival study, immunohistochemistry of liver tissues revealed an increased frequency of focal expression of beta-galactosidase, a marker of senescent cells (*Figure 2E*). Moreover, examination of the epidermis revealed a 50% reduction in thickness in animals lacking Siglec-E (*Figure 2F*). Epidermis tends to thin with increasing age through mechanisms that possibly involve senescent cells (*Lopez-Otin et al., 2013*). Collectively, these data indicate that deletion of Siglec-E results in a faster progression of aging and, consequently, to increased frailty leading to an earlier death.

## Disordered ROS metabolism in Siglec-E$^{-/-}$ mice

Siglec-E regulates inflammatory states upon acute stress (*McMillan et al., 2013*; *Chang et al., 2014*). We speculated that aging might act as a chronic stimulus and investigated whether Siglec-E$^{-/-}$ mice exhibited low-grade signs of inflammation. As noted above, some organs showed accumulation of inflammatory cells (*Figure 2—figure supplement 5*). Inflammation was not due to anti-nuclear antibodies, which are typical of some autoimmune diseases but were undetectable in the sera of Siglec-E$^{-/-}$ and WT mice. To gain mechanistic insights, we analyzed the role of Siglec-E on the management of oxidative stress in innate immune cells. Primary bone marrow neutrophils from Siglec-E$^{-/-}$ mice were more prone to produce oxidative burst upon stimulation, compared to controls (*Figure 3A* and *Figure 3—figure supplement 1*). Additionally, neutrophils lacking Siglec-E secreted higher ROS per cell (*Figure 3B*). Similarly, thioglycollate-recruited peritoneal neutrophils showed a 10% increase in ROS (*Figure 3—figure supplement 2*), corroborating the notion that Siglec-E controls oxidative stress and that the elimination of CD33rSiglec receptors leads to disordered ROS. These observations were also in line with what was shown with a microglial cell line (*Claude et al., 2013*).

Since we found evidence of inflammation in the liver, we used a microarray to examine differential gene expression in this organ in aged Siglec-E$^{-/-}$ animals. The liver is a central organ for the regulation of glucose homeostasis, xenobiotic metabolism and detoxification, and steroid hormone biosynthesis and degradation. Gene expression analysis in aged C57BL/6 mice has indicated that 40% of the genes with changes in expression during aging are associated with inflammation (*Lee et al., 1999*; *Cao et al., 2001*). Another set of genes undergoing changes is related to stress response and chaperones, followed by genes involved in xenobiotic metabolism. Principal component analysis uncovered a subset of genes whose expression differed significantly between genotypes. Pathway analysis of differentially regulated genes suggested changes in leukocyte-mediated inflammation, including increased activation of leukocytes and granulocyte movement, as well as an increase in ROS metabolism in the Siglec-E$^{-/-}$ mice (*Supplementary file 1*). Interestingly, the glutathione S-transferase protein 1 (*gstp1*) gene was found to be down-regulated. Gstp1 catalyzes nucleophilic attack by reduced glutathione on a variety of electrophilic compounds (*Hayes et al., 2005*). The resulting

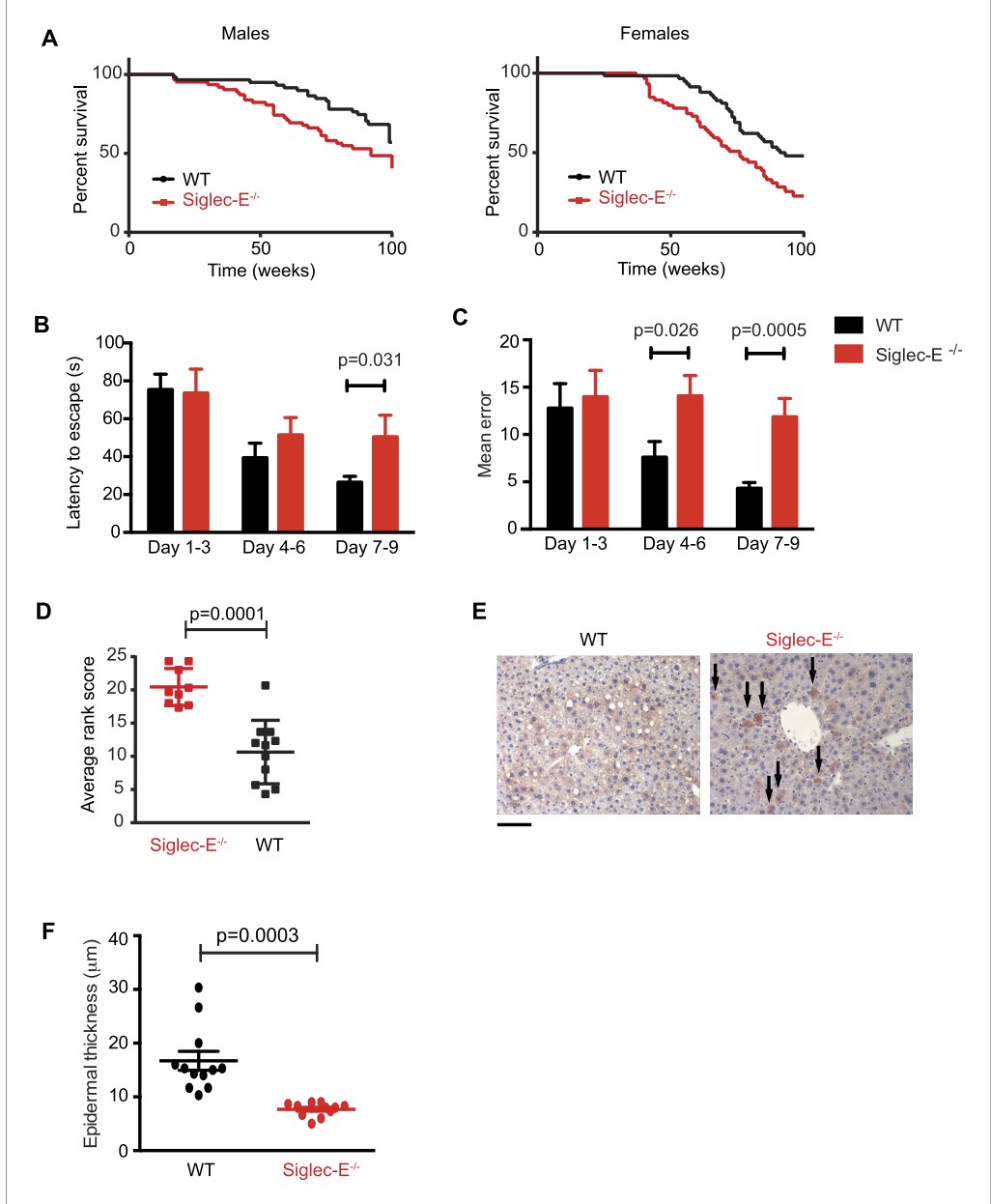

**Figure 2**. Absence of immunomodulatory Siglec-E aggravates aging phenotypes and reduces lifespan in mice. (**A**) Survival curves of WT and Siglec-E$^{-/-}$ male (n = 59–62) and female (n = 58–59) littermates. Data are from two independent cohorts (cohort 1 included 31 WT and 38 Siglec-E$^{-/-}$ males, 31 WT and 33 Siglec-E$^{-/-}$ females. Cohort 2 included 28 WT and 23 Siglec-E$^{-/-}$ males, 27 WT and 26 Siglec-E$^{-/-}$ females). Log-rank test analysis showed significant differences in the survival curves both in males and females (males: $\chi^2$ = 5.833, d.f. = 1 and p = 0.0157; females: $\chi^2$ = 8.821, d.f. = 1 and p = 0.0030). (**B** and **C**) Mice at 80 weeks were assessed for spatial learning and memory via Barnes maze. Latency to escape (**B**) and number of errors before finding the escape box (**C**) are indicated in 3-day interval. Error bars reflect mean ± s.e.m. (n = 11). p was calculated with a Student's *t* test. (**D**) Hair graying of males was evaluated by three independent observers in a blind test. Average rank scores for each mouse are indicated. Error bars reflect mean ± s.e.m. (n = 9–11). p was calculated with a Student's *t* test. (**E**) Surviving mice were sacrificed at 100 weeks of age. Representative field of β-galactosidase staining of liver. Arrows indicate cells with increased localized staining. Scale bar is 100 μm. (**F**) Skin epidermal thickness was measured for WT and Siglec-E$^{-/-}$. Mean and s.e.m. are indicated, n = 12, p was calculated with a Student's *t* test.

The following figure supplements are available for figure 2:

*Figure 2. continued on next page*

*Figure 2. Continued*

**Figure supplement 1**. Expression of Siglec-E in mouse tissues.

**Figure supplement 2**. Exposure to human red blood cell membranes does not impact survival of Siglec-E$^{-/-}$ mice.

**Figure supplement 3**. Deletion of Siglec-E does not alter body weight increase.

**Figure supplement 4**. Hematology and serum chemistry values of male mice at the termination of the study.

**Figure supplement 5**. Absence of Siglec-E is associated with overall increased inflammation in liver and lung.

**Figure supplement 6**. Deletion of Siglec-E does not affect locomotor activity.

complexes are usually less toxic and are eventually metabolized and exported via a glutathione-dependent transport system. In humans, early loss of Gstp1 expression due to promoter hypermethylation results in increased cancer susceptibility (*Lin et al., 2001*). In male mice, Gstp1 is quantitatively the principal glutathione transferase in the liver and is found at lower levels in other organs (*Knight et al., 2007*). In liver from WT male mice, anti-Gstp1 antibodies stained hepatocytes of the zone 1 (*Figure 3C*). A less defined pattern was observed in liver sections of Siglec-E$^{-/-}$ mice. Overall, we observed a substantial difference in staining (*Figure 3—figure supplement 3*). Immunoblot analysis confirmed a 40% reduction in Gstp1 expression (*Figure 3D*). It is interesting to note that *Gsto1* gene, encoding for another glutathione *S*-transferase, was found to be negatively regulated by age in a previous study (*Cao et al., 2001*). Thus, changes in the xenobiotic-metabolizing capacity of the liver appear to be intimately connected to the aging process.

Taken together, these data indicate that absence of Siglec-E leads to a dysregulation of ROS metabolism, resulting in increased levels of reactive species. This phenomenon is due to both an increased production of vacuolar ROS and a deficiency of removal of ROS.

## Siglec-E deficient mice accumulate higher oxidative damage

Many types of ROS that are formed to serve a signaling or protective function can also cause damage spontaneously to lipids, nucleic acids, and proteins. Polyunsaturated fatty acids are a sensitive oxidation targets for ROS because of a damaging chain reaction that takes place once lipid peroxidation is initiated (*Niki, 2009*). DNA bases are also very susceptible to ROS attack, and oxidation of DNA is believed to cause mutations and deletions (*Fraga et al., 1990*). Most amino acids in a protein can be oxidized by ROS, with these modifications leading to a loss of function (*Brennan and Hazen, 2003*). Such damage occurs constantly, and cells must repair it or replace the impaired molecules. Defects that allow oxidative damage to accumulate can contribute to the origin and progression of cancers and neurodegenerative diseases, and in general contribute to the symptoms of aging (*Berlett and Stadtman, 1997*; *Halliwell, 2013*). Similarly, impairment of the processes that control ROS levels can lead to molecular damage. We looked for signs of molecular damage in the organs of the Siglec-E$^{-/-}$ mice, and found a 1.4-fold increase of DNA damage in liver compared to WT (*Figure 4A*). This was in line with the evidence that glutathione S-transferases protect cells against as much as 90% of the damage induced by electrophiles and other free radicals (*Vasieva, 2011*). Brain, spleen, and heart tissues also showed a slight trend towards increase in DNA damage (*Figure 4—figure supplement 1*). Notably, these differences were not detected in the organs of 10-week-old mice (*Figure 4—figure supplement 2*). We then searched for oxidative adducts in proteins elsewhere in the body and found elevated plasma protein-bound 3-nitrotyrosine levels, a marker of protein modification by nitric oxide (NO)-derived oxidants (*Figure 4B*). Similarly, liver of Siglec-E$^{-/-}$ mice showed a trend towards accumulation of oxidized amino acids in proteins compared to WT (*Figure 4—figure supplement 3*). Furthermore, we detected a twofold increase of F2-isoprostanes levels, including 8-iso Prostaglandin F2α and its metabolite 2,3-dinor-8-iso PGF2α in the urine (*Figure 4C,D*). F2-isoprostanes are generated by non-enzymatic peroxidation of arachidonic acid due to free radical species (*Montuschi et al., 2004*). Taken together, these

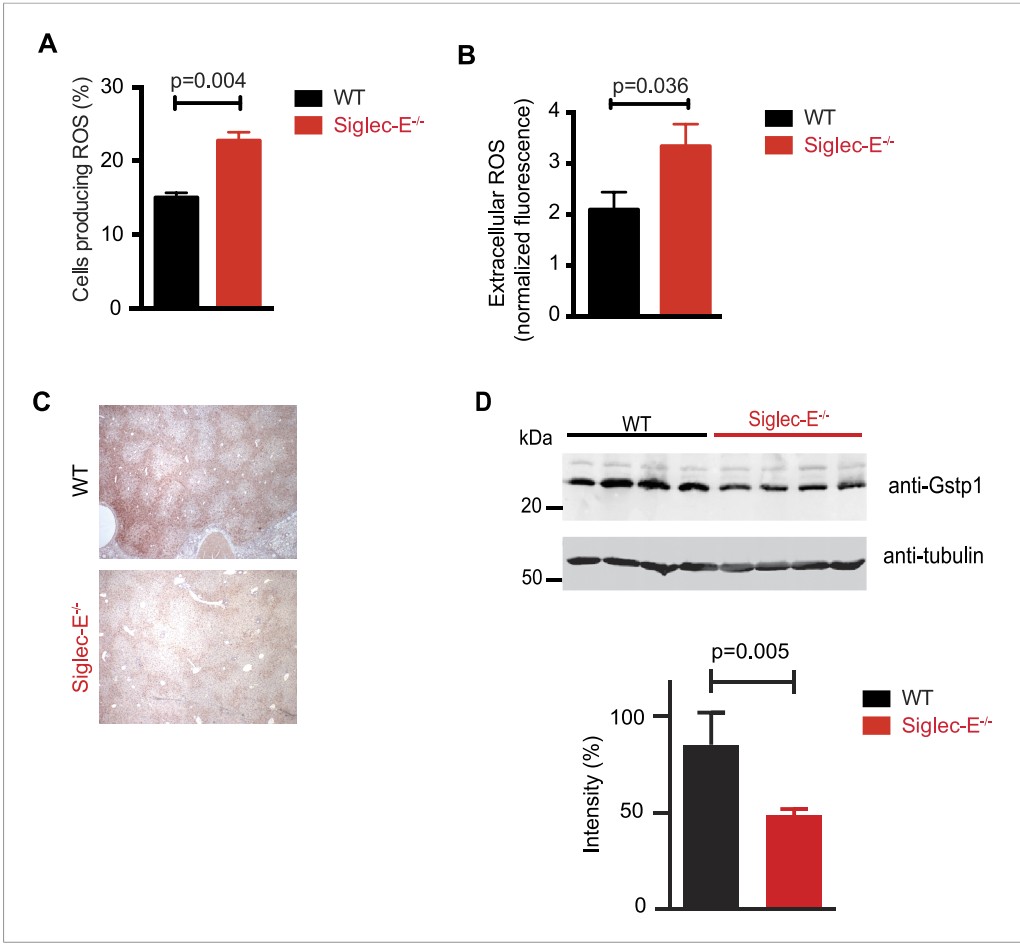

**Figure 3**. Altered ROS homeostasis in mice lacking Siglec-E. (**A**) Neutrophils purified from bone marrow were incubated with immunocomplexes. Cells producing vacuolar ROS were measured by flow cytometry after 60 min. Representative of three experiments, for each n = 3. (**B**) Neutrophils secrete ROS upon stimulation with PMA for 60 min. Extracellular ROS were detected with a probe that does not cross the plasma membrane (n = 11–12). (**C**) Representative Gstp1 immunohistochemistry in liver from WT or Siglec-E$^{-/-}$ male mice at 100 weeks. Expression pattern is altered in the knockout mice. (**D**) Immunoblot analysis and quantification of Gstp1 expression in liver of 100-week-old mice. The level of Gstp1 protein is reduced of about 40%. p was calculated with a Student's *t* test, n = 4.

The following figure supplements are available for figure 3:

**Figure supplement 1**. Neutrophils lacking Siglec-E are more prone to oxidative burst.

**Figure supplement 2**. Thioglycollate-elicited neutrophils from Siglec-E$^{-/-}$ produce higher ROS than WT controls.

**Figure supplement 3**. Gstp1 is found in lower levels in the liver of Siglec-E$^{-/-}$ mice.

data indicate that elimination of Siglec-E leads to accelerated oxidative modification of DNA, proteins and lipids at the systemic level, via elevated ROS and reactive nitrogen species (RNS) production.

## Discussion

CD33rSiglecs differ by a great degree in number, sequence and expression pattern among mammalian species. Together with the evidence that many genes involved in the biosynthesis of sialylated glycoconjugates are rapidly evolving and that some bacterial pathogens can also produce

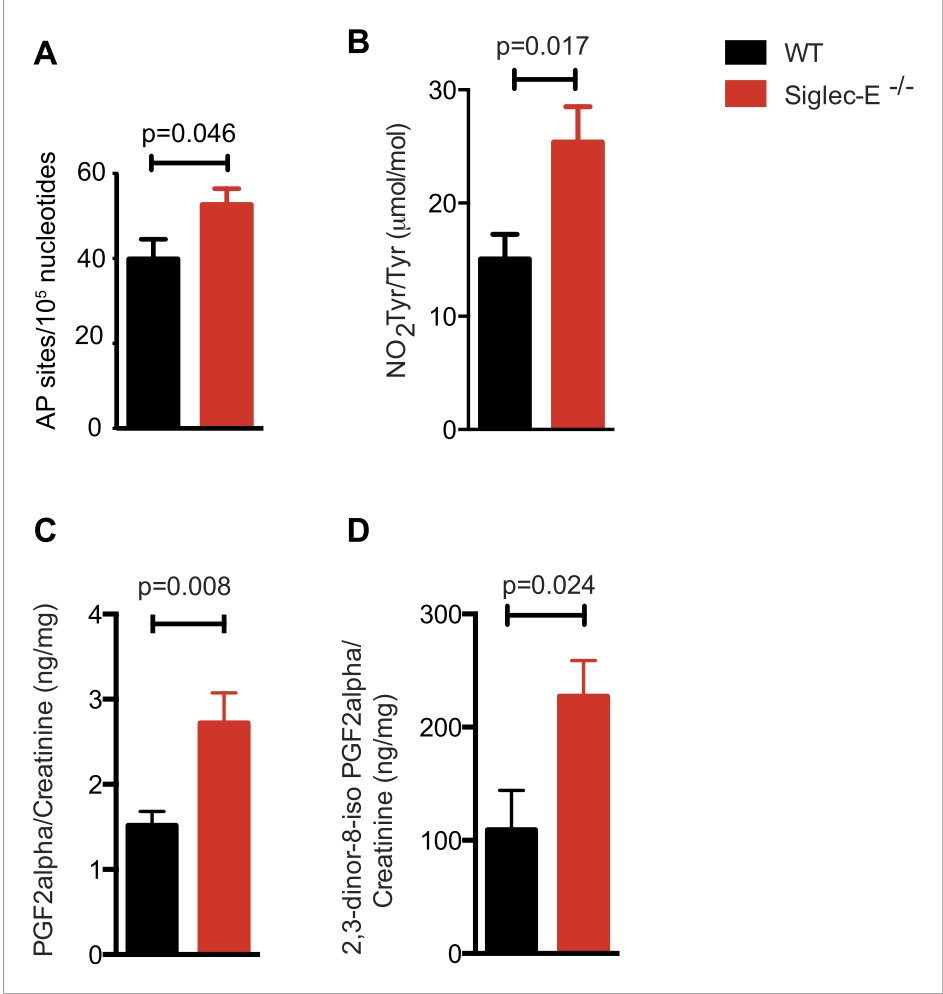

**Figure 4**. Increased oxidative damage in mice lacking Siglec-E. (**A**) Oxidative damage of DNA (AP sites, apurinic/apyrimidinic sites) was measured in the liver from WT and Siglec-E$^{-/-}$ mice at 100 weeks, n = 10. (**B**) Nitrotyrosine (NO$_2$Tyr) accumulation in plasma proteins of Siglec-E$^{-/-}$ mice, n = 8. (**C** and **D**) Concentrations of F2-isoprostanes in urine derived from free radical-induced oxidation of arachidonic acid. Values were normalized by creatinine levels to account for dilution in urine. F2-Isoprostane levels are significantly higher in Siglec-E$^{-/-}$ mice. Data are mean ± s.e.m., Student's *t* test.

The following figure supplements are available for figure 4:

**Figure supplement 1**. Spleen and brain of aged WT and mutant mice have equivalent levels of DNA damage.

**Figure supplement 2**. Young Siglec-E$^{-/-}$ mice exhibit levels of DNA damage comparable to WT.

**Figure supplement 3**. Oxidation of hepatic proteins.

sialylated structures, this led to the hypothesis that CD33rSiglecs function to recognize self in the form of the host sialic acids ('self-sialome'), thereby dampening unwanted responses in the steady state by immune cells, wherein CD33rSiglecs are prominently expressed (*Crocker and Varki, 2001*). Indeed, it has been shown for several CD33rSiglecs that ligation of sialic acid results in the phosphorylation of tyrosine residues of the intracellular immunoreceptor tyrosine-based inhibitory motifs (ITIMs), followed by the recruitment of phosphatases SHP-1 and SHP-2 that turn off the pro-inflammatory cascade (*Angata et al., 2002*; *Ikehara et al., 2004*). Therefore, CD33rSiglecs engage in a dense network of sialic acid-dependent interactions *in cis* on the cell membrane to provide homeostasis.

However, higher affinity ligands on other cells can displace *cis* interactions to take advantage of the inhibitory properties of CD33rSiglecs in these host cells. For instance, many cancer cells produce a heavily sialylated cell surfaces to contact Siglecs of natural killer cells and neutrophils, allowing successful escape from immune recognition (*Hudak et al., 2014*; *Jandus et al., 2014*; *Läubli et al., 2014*). Similarly, bacterial pathogens expressing sialic acids can engage CD33rSiglecs (*Carlin et al., 2009*; *Chang et al., 2014*) and might therefore represent strong driving forces for evolution of this class of receptors (*Varki, 2011*). Of course, other yet unexplored functions of Siglecs might also be associated to their variation in number.

In this study, we analyzed the expansion of *CD33rSIGLEC* genes in the context of evolution of aging. To test our hypothesis that inhibitory CD33rSiglecs might affect aging by regulating ROS homeostasis, we used mice as a simplified model system as deletion of Siglec-E results essentially in the removal of most of the CD33rSiglec receptors from macrophages and neutrophils, which are the main producers of ROS upon inflammatory stimuli. Mice lacking Siglec-E are viable, exhibit no apparent developmental defect and reproduce normally (*McMillan et al., 2013*). Upon acute challenge by lipopolysaccharide-induced airway inflammation or by intravenous administration of bacteria, Siglec-E deficient mice develop exaggerated neutrophil recruitment to the lung and produce higher levels of pro-inflammatory cytokines (*McMillan et al., 2013*; *Chang et al., 2014*). However, as for other Siglec-deficient mice such as CD33 or Siglec-F null animals, no clear phenotype was reported in absence of acute challenge (*Brinkman-Van der Linden et al., 2003*; *Zhang et al., 2007*). Here, we showed that Siglec-E affects ROS homeostasis, and deletion of Siglec-E results in overproduction of ROS. This, together with a secondary impairment in the radical-scavenging enzyme Gstp1 expression leads to higher levels of oxidative adducts of proteins, lipids and DNA, which may lead to acceleration of aging. We thus concluded that Siglec-E impacts aging in mice through regulation of ROS homeostasis. Formally, we cannot state that the upregulation of ROS due to the absence of Siglec-E is the direct cause of the observed molecular damage. In fact, the observed levels of oxidation may be primarily due to the impairment of the detoxification system, which might be controlled upstream by Siglec-E through ROS signaling. Interestingly, the latter hypothesis is in line with a recent revision of the Harman's free radical theory of aging that proposes that ROS generation represents a stress signal to age-dependent damage, rather then being the primary cause of it (*Harman, 1956*; *Hekimi et al., 2011*). However, our findings do support the concept that alteration of the ROS homeostasis accelerates aging. It is also interesting to note that both genetic and pharmacological intervention to reduce ROS levels has produced contrasting results in reverting aging phenotypes, suggesting that too low or too high ROS levels can be equally deleterious. In light of this, it is likely that overexpression of Siglec-E in mice might not result in lifespan expansion. Additionally, this work complements a recent report showing Siglec-E in ROS management in the fibrinogen/β2-integrin signaling pathway (*McMillan et al., 2014*). However, in our assays, bone marrow and peritoneal neutrophils from Siglec-E$^{-/-}$ mice consistently showed a higher ROS production. Lastly, whilst we suggest here that Siglec-E impacts inflammaging by a ROS-mediated mechanism, Siglec-E might also modulate the ability to recognize and remove senescent cells in aging (*van Deursen, 2014*).

Even if the correlation between number of CD33rSiglec family members and lifespan is particularly strong, a higher number of SIGLEC genes may not directly translate in a comparable increase of CD33rSiglec pathway activity. Instead, it is quite possible that the observed gene expansion relates to expression patterns in specific cell types that might ultimately influence cellular CD33rSiglec repertoires. We suggest that specific CD33rSiglec members contribute to the regulation of inflammation by interaction through distinct sialylated structures in vivo, leading to improved regulation of inflammatory responses. Our finding that long-living mammals tend to have more *CD33rSIGLEC* genes is in line with the 'inflammaging' theory (*Franceschi et al., 2000*). Inflammaging postulates that lifetime exposure to antigenic load caused by both clinical and subclinical infections, as well as exposure to noninfective agents, generates low-grade inflammation. This yields additional cytokines and results in a vicious cycle that drives immune system remodeling to a chronic proinflammatory state, ultimately leading to aging and common age-related disorders (*Finch and Crimmins, 2004*; *Finch et al., 2010*). Therefore, age-related diseases may represent the cost of an efficient defense against pathogens conferred by strong inflammation in early life. It also derives that elements that protect against inflammatory damage or mediate repair may have an impact on aging, and that species differences in those elements may translate in distinct patterns of age-dependent disease. The evidence presented in

this work is consistent with a role of CD33rSiglecs in modulating aging derived from chronic inflammation. In fact, CD33rSiglecs are receptors of innate immune cells. Their primary function is to recognize self-associated molecular patterns and modulate host immune responses by regulating cellular reactions, survival, and production of cytokine mediators (*Crocker et al., 2007*; *Chen et al., 2009*; *Cao and Crocker, 2011*; *Varki, 2011*). CD33rSiglecs counteract random molecular damage, which is the main driver of aging. Lastly, *CD33rSIGLEC* gene number correlates with longevity.

In summary, our data provide molecular mechanisms underlying the CD33rSiglec-dependent control of oxidative stress and identify this gene family as modulators of aging pattern and lifespan in mammals.

## Materials and methods

### Animals

Siglec-E$^{-/-}$ mice were described in *McMillan et al. (2013)* and backcrossed with C57BL/6 animals. Heterozygous mice were used to produce WT and Siglec-E$^{-/-}$ littermates control. Mice were housed in cages of groups of 3–5 that did not change from weaning, at $20 \pm 2°C$ under a 12 hr light/12 hr dark photoperiod. Mice were provided with unlimited access to water and to a soy-based chow (Dyets, Inc., AIN-93M, Bethlehem, PA) supplemented with either 0.25 mg/g chow Neu5Gc (by adding purified porcine submaxillary mucin) or 0.25 mg/g chow Neu5Ac (by adding edible bird's nest, Golden Nest Inc., Arcadia, CA). Addition of Neu5Gc or Neu5Ac did not significantly increase the caloric content of the chow. Sterile inflammation was induced via intra-peritoneal injection with 200 µg of human erythrocyte membrane ghosts in 200 µl PBS, along with Freund's complete adjuvant, at an age of 10 weeks. Human erythrocyte membrane ghosts were prepared as described previously (*Hedlund et al., 2008*). A booster injection using Freund's incomplete adjuvant with the same amount of immunogen was given 2 and 4 weeks later. Mice were accessed periodically for evaluation of health status, body weight, and blood tests. Deaths were recorded by animal technicians throughout the study. Decisions for euthanasia of aged mice with severely compromised health were taken by animal technicians, following the guidelines of Institutional Animal Care and Use Committee of the University of California, San Diego, and without involving the scientists.

### Evaluation of hair graying

At approximately 75 weeks of age WT or Siglec-E$^{-/-}$ male mice were ranked blind in order of visible graying to coat fur. In total 26 mice were ranked, including 6 female mice (3 Siglec-E$^{-/-}$ and 3 WT) which had no obvious graying in the coat and therefore acted as a negative baseline. The highest graying was scored 25, and the lowest was scored = 0. The option was available to say that there was no difference between some or all mice, in which case they would be given the same score, but this option was not used by any of the analysts.

### Immunohistochemistry

Organs were extracted from euthanized animals and either fixed in 4% paraformaldehyde (skin, liver, lung, brain) or snap-frozen in OCT and stored at −80°C. Fixed tissues were processed and embedded in paraffin. Paraffin sections were de-paraffinized, blocked, and stained with antibodies following protocols of the UC San Diego Mouse Phenotypic Core http://mousepheno.ucsd.edu/.

### Antibodies

Antibodies were as following: goat anti-Siglec-E (R&D Systems, Minneapolis, MN), rabbit anti-Gstp1 (Sigma–Aldrich, St. Louis, MO), mouse anti-β-actin (Sigma–Aldrich), rabbit anti-β-actin (Cell Signaling, Danvers, MA), anti-CD45 (BD Pharmingen, San Jose, CA), rat anti-Ly6G (clone 1A8, BD Pharmingen), mouse anti-Gstp1 (BD Pharmingen), and rabbit anti-β-galactosidase (Bioss Antibodies, Woburn, MA). Secondary antibodies were from LI-COR (Lincoln, NE) or Jackson ImmunoResearch Laboratories (West Grove, PA).

### Epidermal skin thickness

Dorsal skin was dissected from mice, fixed in 10% paraformaldehyde, and embedded in paraffin. Paraffin sections were prepared at 5-µm thickness and stained with hematoxylin and eosin. Digital photomicrography using the Keyence B6000 (Keyence, Itasca, IL) was performed to collect 400× images, and the epidermal thickness was measured with Keyence BZII Analyzer.

## ROS production

Bone marrow neutrophils were flushed from femur and tibia and purified by Percoll gradient. Peritoneal neutrophils were obtained from peritoneal exudate 16 hr after intraperitoneal injection of 3% thioglycollate. Purity was evaluated by flow cytometry with an anti-Ly6G antibody. For phagosomal ROS, 1 million neutrophils were incubated for 60 min in 700 µl PBS containing 0.5% (wt/vol) glucose and 140 µg/ml Fc OxyBURST (Life Technologies, Grand Island, NY). ROS production was measured with a BD FACScalibur (BD Biosciences). For extracellular ROS, half a million neutrophils were incubated with 10 µg/ml OxyBURST Green $H_2HFF$ BSA (Life Technologies) and phorbol myristate acetate (PMA). ROS production was monitored with a SpectraMax M3 (Molecular Devices, Sunnyvale, CA).

## Microarray analysis

Resected liver samples were placed immediately into RNALater (Qiagen, Valencia, CA) on ice. Tissues were homogenized using a Kinematica homogenizer. RNA was isolated from tissues using RNeasy kit (Qiagen). Concentration and quality of RNA were measured by a NanoDrop ND-1000 spectrophotometer (NanoDrop Technologies) and by a 2100 Bioanalyzer (Agilent). Gene microarray analysis was run by the UC San Diego Biomedical Genomics Microarray Core Facility using a MouseRef-8 v2 Expression BeadChip (Illumina, San Diego, CA). A principal component analysis (PCA) was conducted on the signals obtained from the data matrix (25,697 probes × 6 samples) with Matlab (Mathworks, Inc., Torrance, CA). Data generated from gene array were analyzed for differential geneexpression. Genes were considered differentially expressed with a p value <0.05 (Mann–Whitney U test), and a Log2 fold change of >1 or < −1. The generated list was analyzed using Ingenuity Pathway Analysis.

## Quantification of Gstp1 expression

For immunoblot analysis, liver tissues were washed with PBS and homogenized in RIPA buffer. Cell lysates were spun at 10,000×g. Protein concentration of the supernatant was measured with a BCA kit (Pierce, Rockford, lL). Proteins were run in a SDS-PAGE and transferred to a nitrocellulose membrane. Membranes were incubated with antibodies. Signals were acquired with an Odyssey instrument (LI-COR) and analyzed by Image Studio software (LI-COR).

For immunohistochemistry analysis, liver were fixed in 10% paraformaldehyde and embedded in paraffin. Paraffin sections were prepared at 5-µm thickness and incubated with antibodies. Images were collected using a B6000 microscope (Keyence). Simple image analysis on the brightness (grayscale value of the image) was conducted with the Image Processing Toolbox of Matlab (Mathworks, Inc.). Images were loaded into Matlab, normalized via z score, and then pixel value histograms were inspected. Images were manually inspected for artifact and those pixels were discarded from the analysis (as shown on the images as white lines on far right inset and marked as green in the image to the left). Histograms often showed a bimodal distribution of pixel values indicating a clear demarcation of positive staining. The middle value between this bimodal distribution was used as a criterion to classify between negative and positive staining. Then, the pixels were marked either as bright (red) or dark (blue) and counted. Visual inspection of the classified image was compared to the original image, and if pixels were misclassified, the midpoint was manually changed until the resulting image most clearly separated the tissue differences.

## Measurement of DNA oxidation levels

DNA was extracted from tissues with a DNeasy Blood & Tissue Kit (Qiagen). DNA concentrations of each sample were adjusted to 0.1 µg/ml. The number of apurinic/apyrimidinic (AP) sites was determined using the DNA damage Quantification Kit (Dojindo, Rockville, MD), following the manufacturer's instructions.

## Analysis of oxidative adducts of proteins

At the time of harvest, all tissues were immediately rinsed in ice-cold PBS and frozen at −80°C in PBS containing 100 µM diethylenetriamine pentaacetic acid (DTPA) and 100 µM butylated hydroxytoluene (BHT) in gas-tight containers overlaid with nitrogen. Analysis of oxidative modification of amino acids was done by stable isotope dilution liquid chromatography with on-line tandem mass spectrometry

(LC/MS/MS) using a HPLC interfaced to an AB SCIEX 5000 triple quadrupole mass spectrometer, as described in *Zheng et al. (2004)*.

## Analysis of oxidative adducts of proteins from urine

Urine samples were spun to remove potential cellular debris and then frozen at $-80°C$ until the time of analysis. Urinary creatinine (Cr) levels were quantified on an Abbott Architect machine (Abbott Diagnostics, Abbott Park, IL), according to the manufacturer's instructions. Immediately after thawing, an internal standard ($9α,11α,15S$-trihydroxy-5Z,13E-dien-1-oic-3,3,4,4-d4 acid; PGF2α-d4; Cayman Chemical Company) was added to the sample. Urinary levels of F2-IsoProstanes (PGF2α and 2,3-dinor-PGF2α) were analyzed by stable isotope dilution LC/MS/MS using a HPLC interfaced to an AB SCIEX 5000 triple quadrupole mass spectrometer. To adjust for variations in urinary dilution, the results of F2-IsoProstanes are reported as ratios with urine Cr concentrations.

## Barnes maze test

The Barnes maze test is a spatial learning and memory test originally developed in rats (*Barnes, 1979*), but also adapted for mice (*Bach et al., 1995*). The Barnes maze task has the benefit of minimizing pain and distress to the animal. The Barnes maze apparatus consists of an opaque Plexiglas platform 75 cm in diameter elevated 58 cm above the floor. 20 holes, 5 cm in diameter, are located 5 cm from the perimeter, and a black Plexiglas escape box ($19 \times 8 \times 7$ cm) is placed under one of the holes. Distinct spatial cues are located all around the maze and are kept constant throughout the study. On the first day of testing, a training session was performed, which consists of placing the mouse in the escape box and leaving it there for 5 min. 1 min later, the first trial was started. At the beginning of each trial, the mouse was placed in the middle of the maze in a 10-cm high cylindrical black start chamber. After 10 s the start chamber is removed a bright light is turned on, and the mouse is allowed to explore the maze. The trial ended when the mouse entered the escape tunnel or after 3 min elapsed. When the mouse entered the escape tunnel, it remained there for one minute. When the mouse did not enter the tunnel, it is gently placed in the escape box for one minute. The tunnel was always located underneath the same hole (stable within the spatial environment), which is randomly determined for each mouse. Mice were tested once a day for 9 days. On day 10, a probe test was conducted during which time the escape tunnel was removed and the mouse allowed to freely explore the maze for 3 min. The time spent in each quadrant was determined and the percent time spent in the target quadrant (the one originally containing the escape box) was compared with the average percent time in the other three quadrants. Each session was videotaped and scored by an experimenter blind to the genotype of the mouse. Measures recorded include the number of errors made per session and the strategy employed by the mouse to locate the escape tunnel. Errors were defined as nose pokes and head deflections over any hole that did not have the tunnel beneath it. Search strategies were determined by examining each mouse's daily session and classifying it into one of three operationally defined categories: (1) Random search strategy—localized hole searches separated by crossings through the center of the maze, (2) Serial search strategy—systematic hole searches (every hole or every other hole) in a clockwise or counterclockwise direction, or (3) Spatial search strategy—reaching the escape tunnel with both error and distance (number of holes between the first hole visited and the escape tunnel) scores of less than or equal to 3.

## Locomotor activity

Locomotor activity was measured using an automated monitoring system (Kinder Associates, San Diego, CA). Polycarbonate cage ($42 \times 22 \times 20$ cm) containing a thin layer of bedding material was placed into frames ($25.5 \times 47$ cm) mounted with photocell beams. Each mouse was tested for 120 min.

## Analysis of genomic sequences and gene prediction strategy

Sequences of previously reported human and mouse CD33rSiglecs were retrieved from HGNC (http://www.genenames.org/) and MGI (http://www.informatics.jax.org/), respectively. NCBI annotated CD33rSIGLEC genes from additional mammalian species were used as references for orthologous gene searching. Additional putative CD33rSIGLEC genes were obtained by searching available mammalian genome sequences at UCSC Genome Bioinformatics (http://genome.ucsc.edu/), Ensembl (http://www.ensembl.org/index.html), and NCBI (http://www.ncbi.nlm.nih.gov/gene). As SIGLEC genes contain introns, we adopted and modified a previously established search strategy (*Shi and Zhang, 2006*). First, we used BLAT/TBlastN to identify the genomic location of a putative

CD33rSiglec gene in a genome with a previously reported CD33rSiglec as a query. Secondly, Genscan was used to predict the gene structure found in this genome location. Simultaneously, the genomic DNA sequences of the putative CD33rSIGLEC gene and the known CD33rSiglec protein sequence were used to conduct a protein-to-genomic sequence alignment by Wise2. Furthermore, to ensure the accurate prediction of a CD33rSIGLEC, the obtained putative protein sequence was examined by TMHMM V.2.0 or SPLIT 4.0 SERVER for the presence of a transmembrane domain and examined by SignalP 3.0 Server for the presence of a signal peptide. Additional domain evaluation was also conducted in Pfam 25.0 (http://pfam.sanger.ac.uk/) to find the existence of V-set and C2-set domains in the putative CD33rSiglec-encoding gene. All candidates then underwent BLAST analysis against the entire GenBank to ensure that their best hits are annotated as CD33rSiglecs. This step is important because CD33rSIGLECs are known to be related to other SIGLEC genes (e.g., CD22, MAG, and SIGLEC15), as well as other cell surface Ig-like receptors. The above gene search strategy was also applied for predicting all of the KLKs, TLRs, and IgG Fc receptors in mammals under consideration, though the criteria used in gene structure evaluation were gene family dependent.

## Definition of functional genes

Based on previous studies on CD33rSiglecs some particular characteristics are considered in order to define a gene as encoding a functional Siglec (*Crocker et al., 1998*). One criterion is that a Siglec protein is capable of binding sialylated glycans. This binding activity requires a conserved arginine residue in the Ig-like V-set domain. The other criterion is that a functional Siglec protein should contain either a cytosolic tail with at least one ITIM motif or a transmembrane domain carrying a positively charged amino acid. The eventually acquired candidate CD33rSiglecs in each species were considered as true orthologs and used in our correlation analysis. Defining a functional gene using our gene prediction approach is not black and white, due to the nature of incomplete genome sequences or genome sequencing errors. Thus, a few rules were considered during our prediction process. First, when entire exons of a gene (usually one or two) are missing due to a gap in the genome but ORFs remain undisrupted in the available sequences, we treat the case as a functional gene. Second, different species have variable quality of genome coverage. For example, human and mouse genomes have the highest coverage (>12x) out of all mammals, whereas cat and pig genomes have the lowest ones (<5x). Notably, we did not see a trend of higher genome coverage leading to more CD33rSiglec-encoding genes. Moreover, even when we focused only on the species with comparable genome coverage (opossum, dog, marmoset, cow, rhesus macaque, orangutan, chimpanzee, elephant, horse, and rat), the correlation of the number of CD33rSiglecs and maximum life span was still maintained. Therefore, the quality of genome sequencing in mammalian species likely had no impact on our overall findings and conclusion. Finally, we also observed genome sequencing errors in the form of 1 bp mutations or indels in two KLK genes, one IgG Fc receptor gene, and five TLR genes. In this study, such sequences were also considered as functional genes in all species. Notably, the number of TLR genes predicted in several mammalian species using our approach is equal to those reported earlier (*Leulier and Lemaitre, 2008*).

## Longevity and body weight data

Data regarding maximum lifespan and average adult body weight for mammalian species are from AnAge: the animal aging and longevity (http://genomics.senescence.info/species/) (*de Magalhães and Costa, 2009*).

## Statistical analysis

Unpaired Student's t-test was used for comparisons involving two groups. Lifespan analysis was performed using log-rank (Mantel–Cox) test. Median survival refers the time at which half the subjects have died. The Pearson's coefficient was used to calculate correlation. All variables except the gene number counts were log-transformed for statistical analyses. Prism 6 Program (GraphPad, La Jolla, CA) was used for most of the statistical analyses. PGLS and FIC analysis were conducted in COMPARE 4.6b (http://www.indiana.edu/~martinsl/compare/) using a degree of freedom of 11, with three (one for calculating contrast and two for estimating the slope and the intercept) subtracted from 14 (the total number of taxa). Phylogenetic regressions controlled for the body mass were run using pglmEstLambda in the CAIC package (Comparative Analysis of Independent Contrasts) in R. The function of pglmEstLambda uses the PGLS method, estimating $\lambda$ as an index of the strength of the phylogenetic pattern in the data. The model included CD33rSiglec gene numbers as response, maximum lifespan and body mass as covariates. For $\lambda$ values, we followed the rationale described in *Navarrete et al. (2011)*.

## Study approval

All animal studies were approved by the IACUC of the University of California San Diego.

## Accession number

Gene expression data are available at the GEO Archive (GSE64760).

## Acknowledgements

We thank Ana Navarrete for help with the CAIC program, Shoib Siddiqui for help with the preparation of peritoneal neutrophils, and all the members of the Varki and Gagneux laboratories for discussion. This work was supported by grants from the Ellison Foundation, the NIH (P01 HL107150 to AV, P01 HL076491 to SLH, and RO11065732 to PG), the Mathers Foundation of New York, a fellowship from the Novartis Foundation for medical-biological research (to FS), and a Samuel and Ruth Engelberg Fellowship from the Cancer Research Institute (to OMTP). Mass spectrometry studies were performed in a facility partially supported by a Center of Innovation Award from AB SCIEX.

## Additional information

### Competing interests

CKG: Reviewing editor, *eLife*. The other authors declare that no competing interests exist.

### Funding

| Funder | Grant reference | Author |
|---|---|---|
| National Institutes of Health (NIH) | P01 HL107150 | Ajit Varki |
| National Institutes of Health (NIH) | P01 HL076491 | Stanley L Hazen |
| National Institutes of Health (NIH) | RO11065732 | Pascal Gagneux |
| National Institute of Health (NIH) | K99HL123485 | Casey E Romanoski |
| G Harold and Leila Y. Mathers Foundation | | Ajit Varki |
| The Ellison Foundation | | Ajit Varki |
| Novartis Stiftung für Medizinisch-Biologische Forschung | Post-doctoral Fellowship | Flavio Schwarz |
| Cancer Research Institute (CRI) | Samuel and Ruth Engelberg Fellowship | Oliver MT Pearce |

The funders had no role in study design, data collection and interpretation, or the decision to submit the work for publication.

### Author contributions

FS, OMTP, XW, Conception and design, Acquisition of data, Analysis and interpretation of data, Drafting and revising the article; ANS, HL, CH, Acquisition of data, Analysis and interpretation of data, Drafting and revising the article; NV, Acquisition of data, Analysis and interpretation of data, Revising the article; JOG, Analysis and interpretation of data, Drafting the article, Final approval of the version to be published; CER, Analysis and interpretation of data, Critical revision of the article, Final approval of the version to be published; HLin, XF, AG-B, PS, Acquisition of data, Critical revision of the article, Final approval of the version to be published; CKG, SLH, Conception and design, Analysis and interpretation of data, Revising the article; AV, PG, Conception and design, Analysis and interpretation of data, Drafting and revising the article

### Author ORCIDs

Oliver MT Pearce, http://orcid.org/0000-0003-3953-1629

## Ethics

Animal experimentation: All animal experiments were approved by the IACUC of the University of California San Diego (protocol S01227).

# Additional files

## Supplementary file

• Supplementary file 1. Pathway analysis, calculated with Ingenuity Pathway Analysis software. The p-value is a measure of the likelihood genes in the process appear by chance and it was calculated by right-tailed fisher exact test. The z-score is calculated by IPA software algorithm, and predicts the change in the biological function direction. A z-score $\geq 2$ or $\leq -2$ is considered significant.

## Major dataset

The following dataset was generated:

| Author(s) | Year | Dataset title | Dataset ID and/or URL | Database, license, and accessibility information |
|---|---|---|---|---|
| Pearce OMT, Varki A | 2013 | GEO_GA_illumina_ expression_Oliver Pearce | http://www.ncbi.nlm.nih.gov/ geo/query/acc.cgi? acc=GSE64760 | Publicly available at the NCBI Gene Expression Omnibus (GSE64760). |

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
