## [Decision Letter]

Thank you for sending your work entitled “Siglec Receptors Impact Mammalian Lifespan by Modulating Oxidative Stress” for consideration at *eLife*. Your article has been favorably evaluated by Tadatsugu Taniguchi (Senior editor) and two reviewers, one of whom is a member of our Board of Reviewing Editors.

The Reviewing editor and the other reviewer discussed their comments before we reached this decision, and the Reviewing editor has assembled the following comments to help you prepare a revised submission.

This study reports interesting findings that implicate Siglec-E in mammalian aging. The authors find accelerated aging and reduced life span in Siglec-E deficient mice. Their analysis suggests that lack of Siglec-E leads to increased ROS accumulation and ROS mediated damage in several tissues. Overall, the interpretation of the data is that the lack of Siglec-E results in increased inflammatory tone with subsequent increase in ROS production and ROS-mediated damage that in turn contributes to accelerated aging.

I think this is an interesting study that is suitable for publication in *eLife* because it reports an unexpected finding that is of broad interest. Also, the authors provide a reasonable level of mechanistic analysis of the aging phenotype in these animals.

*Reviewer #1*:

1) The authors observe accumulation of senescent cells in Siglec-E KO mice. It seems to me that this may be the key aspect of the mechanism here. Accumulation of senescent cells has been recently shown to contribute to aging phenotype in mice (Nature, 2011, November 2, 479(7372): 232-236). One obvious question to ask is whether Siglec-E is directly involved in the control of phagocytosis of senescent cells. Siglec would generally be expected to inhibit phagocytosis, but this may only be true for inflammatory phagocytosis. In contrast, homeostatic phagocytosis of senescent cells may be facilitated by Siglec-E, if senescent cells have altered expression of sialic acids on their surface glycoproteins. It would be worth investigating this possibility as it may provide a much broader insight into aging.

2) The part of the study about correlation of *CD33* gene number with maximal life span in different mammalian species is interesting, but clearly very tenuous. Presence of multiple paralogs of *CD33* gene may or may not result in increased 'CD33 pathway' activity. It may, for example, relate to cell type specific expression. The authors should comment on that. Without qualifying statements, this part takes away a bit from the rest of the paper.

3) The authors should expand the overview of the CD33 family in the Introduction. As this paper is intended for a broad audience, including investigators not familiar with innate immunity and Siglec biology, it would be very useful to give a brief overview of CD33 family in mice and humans, their expression patterns and known specificities.

*Reviewer #2*:

This will be an important publication when a major gap is addressed in the comparative genomics: short lived primates were omitted for which sequence data are available. The other concerns are minor.

1) Introduction: Please correct sweeping statements that distract from merits of this study. Particular examples are:

a) The initial sentence is too broad. Some species have indeterminate lifespans without evident senescence, e.g. the naked mole-rat and many bivalve species.

b) How are Siglecs typically expressed on leukocytes when they are found in most if not all cells?

c) How is inflammation and important determinant of aging? Every damaged tissue has some level of inflammatory process, but that is not to make inflammation a determinant in any specific sense.

2) The comparative genomics of Siglecs for 15 species includes 4 primates besides human, but must include shorter lived primates for which complete DNA sequences are available, particularly the common marmoset and mouse lemur, with 12 year lifespan. Other lemur genomes may be available. The maximum lifespans are based on hugely different population samples, e.g. human >5 billion, bonobo <1000 in captivity. It would be better to use a sample-size adjusted maximum lifespan for humans.

3) Findings on Siglec-E^−/−^ mice are very interesting:

a) Please give body weight of adults.

b) In text, give effect size of significant changes, it is not necessary to state in text if statistical significance was reached, that is shown in figure legends and tables.

c) For microarray on aging liver, please compare these findings to the extensive microarray data on aging wildtype mouse liver from Weindruch, Prolla, and others.

---

## [Author Response]

Reviewer #1:

*1) The authors observe accumulation of senescent cells in Siglec-E KO mice. It seems to me that this may be the key aspect of the mechanism here. Accumulation of senescent cells has been recently shown to contribute to aging phenotype in mice (Nature, 2011, November 2, 479(7372): 232-236). One obvious question to ask is whether Siglec-E is directly involved in the control of phagocytosis of senescent cells. Siglec would generally be expected to inhibit phagocytosis, but this may only be true for inflammatory phagocytosis. In contrast, homeostatic phagocytosis of senescent cells may be facilitated by Siglec-E, if senescent cells have altered expression of sialic acids on their surface glycoproteins. It would be worth investigating this possibility as it may provide a much broader insight into aging*.

This is a very interesting suggestion. In a previous study, we found that resident peritoneal macrophages, which express detectable levels of Siglec-E, showed a M2 polarization (high CD206 and low CCR2 expression) compared with macrophages from WT control mice when co-cultured with tumor cells for 2 days (Läubli, et al., Proc Natl Acad Sci U.S.A., 111:14,211-14,216, 2014). This suggests a skewing of Siglec-E^−/−^ macrophages to an M2 phenotype, although it is not clear if this would impact the ability of the macrophage to phagocytose. Furthermore, glycan patterns are suggested to change during aging. This phenomenon would also contribute to the phagocytic activity (Dall'Olio F et al., Ageing Res Rev., 2013 March, 12(2):685-98). In our study presented here we have described increased ROS production as one potential mechanism to explain the inflammaging phenotype seen. While we appreciate that Siglec-E might modulate phagocytosis of senescent cells, we respectfully suggest that a full exploration of this phenomenon will involve a lot of effort, and is beyond the scope of this current manuscript. We have added the following statement to the Discussion to include the suggestion of increased phagocytosis as another potential mechanism:

“Lastly, whilst we suggest here that Siglec-E impacts inflammaging by a ROS-mediated mechanism, Siglec-E might also modulate the ability to recognize and remove senescent cells in aging (van Deursen JM1, Nature, 2014 May, 22;509(7501):439-46)”.

*2) The part of the study about correlation of* CD33 *gene number with maximal life span in different mammalian species is interesting, but clearly very tenuous. Presence of multiple paralogs of* CD33 *gene may or may not result in increased 'CD33 pathway' activity. It may, for example, relate to cell type specific expression. The authors should comment on that. Without qualifying statements, this part takes away a bit from the rest of the paper*.

We fully agree with the reviewer. In this manuscript, we never claim that species with fewer *SIGLEC* genes would have fewer Siglec molecules on their cells, with lower Siglec function, nor that having more genes directly translates in higher pathway activity. Also, Siglecs exhibit distinct binding specificities, and different organisms might express different patterns of sialylated ligands; this aspect should be taken into account when measuring Siglec pathway activity. Instead, our view is that Siglecs contribute to the control of inflammation, and these control mechanisms are more elaborate in species with a higher number of genes, resulting in a better management of inflammatory responses and lower molecular damage. At the same time, we felt important to include these data in the current manuscript as the correlation was notably strong, even after adequate corrections for phylogeny and body mass. We believe that the additions suggested by reviewer #2 further strengthen the case.

We have added this text in the Discussion to further explain this point:

“Even if the correlation between number of CD33rSiglec family members and lifespan is particularly strong […] by interaction through distinct sialylated structures in vivo, leading to improved regulation of inflammatory responses”.

*3) The authors should expand the overview of the CD33 family in the Introduction. As this paper is intended for a broad audience, including investigators not familiar with innate immunity and Siglec biology, it would be very useful to give a brief overview of CD33 family in mice and humans, their expression patterns and known specificities*.

We have added additional information about CD33rSiglec expression pattern and known specificities, and general references as appropriate. We have added the following text to the Introduction:

“CD33rSiglecs in humans are numbered (e.g., Siglecs-3, -5, -6, -7, -8, -9, -10, −11, -XII, −14 and −16), while murine CD33rSiglecs […] it appears that each Siglec has unique sialoglycan specificity profile with regard to the type of sialic acid, its linkage and the composition of underlying glycan structure”.

Reviewer #2:

*1) Introduction: Please correct sweeping statements that distract from merits of this study*.

We have either deleted or modified the text to make it more specific as suggested. Siglecs are expressed mainly on cells of the hematopoietic lineage, in a species-specific fashion, and in brain microglia, and are not widely expressed on all cell types. However, there are some specific Siglec members that have been found on human epithelial cells (Siglec-XII) or on human amnion (Siglec-5 and -14).

*2) The comparative genomics of Siglecs for 15 species includes 4 primates besides human, but must include shorter lived primates for which complete DNA sequences are available, particularly the common marmoset and mouse lemur, with 12 year lifespan. Other lemur genomes may be available. The maximum lifespans are based on hugely different population samples, e.g. human >5 billion, bonobo <1000 in captivity. It would be better to use a sample-size adjusted maximum lifespan for humans*.

With regard to adding more genomes to the correlation, we do agree that this is a very valuable exercise. Data from the common marmoset (or white-tufted-ear marmoset) were already included in the previous version of the manuscript, in the “Species” column of Figure 1—figure supplement 1 we wrongly indicated the species *Callithrix geoffreyi*, whereas the data referred to the common marmoset *Callithrix jaccus*. We corrected this error in Figure 1—figure supplement 1, as appropriate. Notably, when we added the *CD33rSIGLEC* gene numbers from the genomes of three other short-lived primates (*Saimiri boliviensis*, *Tarsius syrichta* and *Otolemur garnettii*), the overall correlation remained strong (R^2^ = 0.661 in logarithmic scale, R^2^ = 0.752 in linear scale). Unfortunately, we were not able to accurately determine the number of *CD33rSIGLEC* genes from the mouse lemur genome, due to the poor quality annotation of this genome.

As the reviewer notes, the maximum longevity of humans is based on a considerably larger sample, compared to other species. The AnAge database mentions that a human maximum longevity 90 or 100 years might be more adequate for correlative studies. Therefore, we re-evaluated the correlation using a maximum longevity value of 90 years for humans, as in previous studies. Notably, the overall correlation remained strong (R^2^ = 0.649 in the logarithmic scale, R^2^ = 0.843 in the linear scale).

Since these additions did not change the primary correlation, we would like to request the following compromise. The primary data and presentation remains unchanged, however, we have added the following to the Results section:

“Since the time that these data were originally collected and evaluated, additional genome sequences have become available. […] Notably, the overall correlation between number of *CD33rSIGLEC* genes and maximum longevity remained strong (R^2^ = 0.649 in the logarithmic scale, R^2^ = 0.843 in the linear scale).”

*3) Findings on Siglec-E*^−/−^
*mice are very interesting*:

*a) Please give body weight of adults*.

Data on body weight are indicated in Figure 2—figure supplement 3.

*b) In text, give effect size of significant changes*, *it is not necessary to state in text if statistical significance was reached*, *that is shown in figure legends and tables*.

We have amended the text and the figure legends accordingly.

*c) For microarray on aging liver, please compare these findings to the extensive microarray data on aging wildtype mouse liver from Weindruch, Prolla, and others*.

We compared the gene expression profiles with those reported in the literature. We added the following wording to the Results section of the revised manuscript:

“The liver is a central organ for the regulation of glucose homeostasis, xenobiotic metabolism and detoxification […]. Another set of genes undergoing changes is related to stress response and chaperones, followed by genes involved in xenobiotic metabolism”.

“It is interesting to note that *Gsto1* gene, encoding for another glutathione S-transferase, was found to be negatively regulated by age in a previous study (Cao et al.. Genomic profiling of short- and long-term caloric restriction effects in the liver of aging mice. Proc Natl Acad Sci U S A. 2001 September 11;98(19):10,630-5). Thus, changes in the xenobiotic-metabolizing capacity of the liver appear to be intimately connected to the aging process”.